# Climate-Related Co-Benefits and the Case of Swedish Policy

**Mikael Karlsson** [1,*] **, Nils Westling** [2] **and Oskar Lindgren** [1]

1 Department of Earth Sciences, Uppsala University, 752 36 Uppsala, Sweden
2 Swedish Energy Agency, 631 04 Eskilstuna, Sweden
* Correspondence: mikael.karlsson@geo.uu.se

**Abstract:** There is strong scientific evidence for the existence of the significant economic value of several climate-related co-benefits. However, these are seldom recognised in policy-making, and knowledge is still scarce on some co-benefit types and categories. To identify research needs and highlight policy-making opportunities, we propose a new framework and three-type-taxonomy of climate-related co-benefits. We define climate policy co-benefits, such as improved air quality, as 'Type 1'; co-benefits for climate objectives from policy-making in other fields, such as taxation, as 'Type 2'; and co-benefits from policies designed to achieve multiple objectives as 'Type 3'. In order to apply the framework and to analyse how co-benefits have been regarded in policy-making in a climate pioneering country, we also explore the case of Sweden. It is shown that several co-benefits exist, but that these are overlooked almost entirely in policy-making, constituting a bias against climate mitigation. In order to counteract this problem, the article presents a number of recommendations, including a call to researchers to identify and quantify additional co-benefits and to policy-makers on governance reforms, including the need to organise policy-making processes and set decision criteria that promote the consideration of co-benefits.

**Keywords:** co-benefit; climate mitigation; climate policy; impact assessment; policy-making

## 1. Introduction

A recent comprehensive review of peer-reviewed scientific journals showed that the co-benefits of climate policy are often of significant economic value, but also that such co-benefits are insufficiently researched and seldom considered in policy-making [1]. The review concluded that an enhanced knowledge and awareness of such climate policy-related co-benefits can prompt stricter policies, and other research has shown that knowledge of climate policy-related co-benefits could counteract climate science denial [2]. Moreover, important co-benefits can also occur on the opposite side of the coin, when policies and measures in other fields benefit climate objectives, but such synergies are even less studied and promoted [3]. The relevance of climate-related co-benefits has been considered significant in previous studies [1,4–6], and includes improved air quality and diets—which save millions of lives worldwide—as well as reduced fuel poverty and improved energy security.

In its sixth assessment report on climate mitigation, the Intergovernmental Panel on Climate Change (IPCC) highlighted this growing scientific evidence, recognizing that policy packages that steer towards climate mitigation would benefit from not only looking at climate benefits but also at co-benefits [7]. However, despite being referred to and argued for more often, co-benefits are rarely measured or quantified, and even less so included in policy-making frameworks and processes [1,4,7]. To ensure that all relevant aspects are considered in policy-making, these various types of climate-related co-benefits deserve more attention by scholars, politicians and civil servants. Considering the need for integrated and coherent policies, it would also be desirable if these groups would pay more attention to measures that promote several policy objectives simultaneously. However, given the width of potential co-benefits in multiple policy fields, different interpretations

of co-benefits, and lack of an agreed-upon taxonomy or framework for co-benefits, it is difficult today for policy-analysts and policy-makers to identify and include co-benefits of different types in decision-making.

In this article, we aim to improve clarity, highlight research needs, and promote policy dialogue and development regarding climate-related co-benefits by exploring and elaborating a novel framework that includes a taxonomy with three different types of co-benefits. More specifically, we ask (i) what the scientific literature says about different types of climate-related co-benefits, and (ii) how such co-benefits are or could be identified and considered in policy-making. In order to provide an in-depth analysis of public policy-making barriers and opportunities in this context, we analyse the situation in Sweden, a country long considered to be a climate policy pioneer [8–10]. The recommendations we present are intended to stimulate further studies and dialogue that ultimately may foster policy-making that enhances synergies and consequently improves the possibility of achieving climate objectives in parallel with other societal goals.

In the following, we first introduce and illustrate the occurrence and scientific knowledge of climate-related co-benefits. We present which types and categories of co-benefits that exist based on a new, straightforward and user-friendly framework, and analyse how co-benefits are conceptualised and categorised in academic studies. Second, we provide the results of an empirical study on Swedish public policy linked to various types of climate-related co-benefits in the framework and describe how these have been considered in concrete policy-making. The article thereafter concludes with a discussion that includes a number of recommendations relevant for both research and policy-making.

## 2. Materials and Methods

This article comprises a general and case-specific part, followed by a common discussion with recommendations for both science and policy-making. The first and general part focuses on the occurrence of various types of climate-related co-benefits. These are identified, described and elaborated on through an inductive qualitative exploratory approach, which is suitable when knowledge on a topic is scarce and needs to be categorised and structured [11]. In this case, this leads to the establishment of a taxonomy and framework intended to help point out research needs and structure future studies, as well as to stimulate policy development. The explored and analysed material mainly consists of scientific studies on climate-related co-benefits found in reviews and articles in scientific journals.

The second part is a case study focused on Swedish policy, where the framework is applied in a concrete setting. The case study approach is well-recognized in the social sciences as a means to generate novel and in-depth knowledge of specific issues [12]. Although at times criticized for their limited capacity for broader generalizations [13,14], case studies are oft-used and well suited to understand complex issues, generate policy-relevant knowledge and address research requiring contextual analysis [15].

In line with the two main types of co-benefits outlined in the framework, we have studied both climate policy making and policy making in another central area—general taxation. The exploration focuses on the preparatory policy process taking place before final policy decisions are made. The Swedish Public Inquiry System—in which policy proposals are investigated thoroughly—plays a central role in Swedish policy-making. These inquiries therefore serve as a good empirical basis for analysing the extent to which co-benefits are considered in Swedish policy making, and constitute the main part of our analysis. The more specific focus is outlined in Section 4.

We have chosen 1990 as a starting point because IPCC at that time released its first assessment report [16] and because the Swedish government at that time proposed a tax on carbon dioxide emissions that passed through the parliament [17], implying an early recognition in Swedish politics of the importance of climate policy. To include the most recent full term of office and the most recent inquiries, the election day in September 2022 was chosen as an endpoint, after which no thorough policy process at the time of writing has been carried out. We have decided to focus on co-benefits from climate policies

and co-benefits for climate objectives from other policy fields, since these synergies are comparatively well-studied in science—especially the former. Co-benefits from policies designed to achieve multiple objectives are less explored in research, and contemporary policy making is insufficiently focused on describing, and even less so on calculating or quantifying, such synergies. We elaborate further on the complexities of such co-benefits in the discussion.

## 3. Climate-Related Co-Benefits

This section begins by providing an overview of the co-benefits in various policy areas resulting from climate policy and emission mitigation measures. Next, an outline is given of co-benefits in terms of greenhouse gas emission reductions that follow policies and measures in other areas. It should, however, be noted that policy objectives are not always articulated or distinctly formulated. This is due to a number of factors, including vague goal formulations and the complexity of certain issues [18]. It is also not always clear who defines the primary or ancillary outcomes. A policy described by some as promoting greenhouse gas mitigation may be regarded by others as an air quality policy, and vice versa. To avoid confusion about intent and outcomes, we have as much as possible deciphered how the various scrutinized policies have been presented.

### 3.1. Climate Policy Co-Benefits

The co-benefits of policies and measures aimed to decrease greenhouse gas emissions—hereafter termed "climate policy co-benefits"—were already pointed out in the academic literature three decades ago [19], but such benefits have seldom been adequately incorporated into the economic analysis of climate change policy—for example in cost-benefit assessments, which inform decision-making [20–23]. Since then, the number of published scientific articles on co-benefits has increased significantly [1] (see Figure 1).

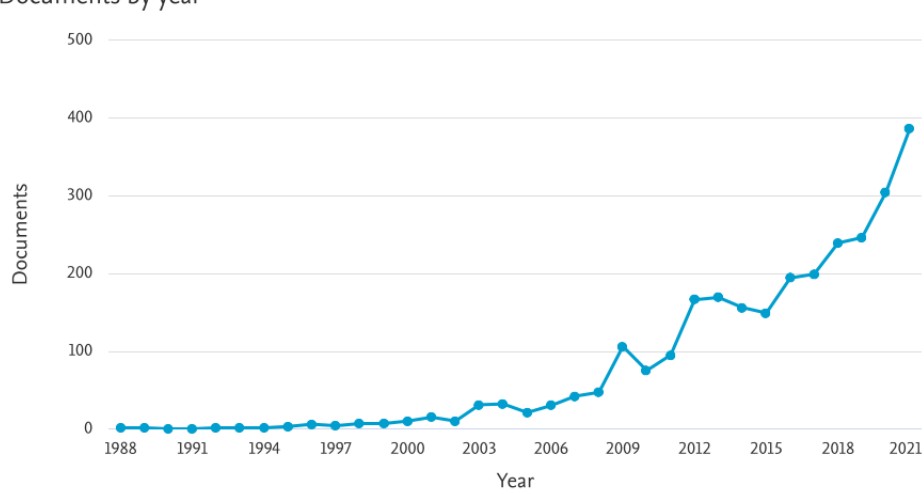

**Figure 1.** Number of scientific articles on climate policy co-benefits in 1988–2021. Source: SCOPUS, 30 November 2022 (see Supplementary Material).

While terms such as 'ancillary benefits', 'double dividend' and 'win-win' were previously commonly used, the 'co-benefit' concept has become mainstream in the academic literature and is defined by the IPCC as "a positive effect that a policy or measure aimed at one objective has on another objective, thereby increasing the total benefit to society or the environment" [24].

A review by Karlsson et al. [1] that covered 239 peer-reviewed scientific journal articles found that a broad set of co-benefit categories has been targeted by researchers. From the start, most studies on climate policy co-benefits have focused on improved air quality resulting from fossil fuel phase-out, for example reduced emissions of particulate matter

and nitrogen oxides—pollutants that can seriously impair public health [25]. In recent years, the study of co-benefits has broadened, including the following categories:

- Diet: e.g., transitioning towards eating less meat and more plant-based alternatives could reduce global mortality by up to 10% while simultaneously reducing greenhouse gas emissions by 29–70% by 2050 [26].
- Physical activity: e.g., increased walking and cycling in France could reduce greenhouse gas emissions while simultaneously saving EUR 749 billion in health costs up until 2050 [27].
- Soil and water quality: e.g., increased soil carbon pool could sequester 50 ppm atmospheric carbon dioxide by 2100 while increasing agricultural productivity [28].
- Biodiversity: e.g., reduced deforestation in so-called REDD+ countries could reduce global species extinction projections by up to 93%, while sequestering carbon, compared to a business-as-usual scenario [29].
- Energy security: e.g., energy efficiency improvements in India could lead to improved energy security since such measures could reduce oil imports by 266 Million barrels of oil equivalents by 2035 [30].
- Economic performance: e.g., implementation of a carbon tax that could reduce emissions by 75% in 2050 in Mexico would have positive effects on GDP [31].

A large proportion of these climate policy co-benefit articles in the literature employ quantitative methods, but only a fraction present monetary valuations [1]. This is unfortunate, since establishing a cost–benefit analysis (CBA) is often required in public climate policy-making despite substantial criticism against using CBAs for complex issues such as climate change [22]. In these situations, a particularly useful estimate of co-benefits is USD/tCO2e, since it allows clear comparisons with, for example, the social cost of carbon, the cost of greenhouse gas mitigation in a specific sector, and carbon pricing mechanisms. However, even fewer studies provide such estimates, with most of them relating to the health benefits of improved air quality [1].

Despite the substantial evidence of health co-benefits from improved air quality and the emerging evidence of other climate policy co-benefits, this knowledge is seldom applied in climate policy-making in general. To what extent this claim is valid in Swedish policy-making will be explored in Section 3.

### 3.2. Climate Co-Benefits

Policies and measures in fields other than climate policy can also bring about benefits for climate objectives, such as greenhouse gas emissions reductions. We term these "climate co-benefits". Some evidence of climate co-benefits can be identified in the scientific literature, but the number of studies is limited [1]. In the following, we present a few studies to illustrate the diversity of the potential cases.

Rao and Min [32] explored the possible climate impacts of reduced economic inequality between and within countries, and found that greenhouse gas emissions would most likely be reduced with increasing global equality. The study showed that energy intensity reductions from income growth in emerging economies can decrease emissions intensity significantly, and that reduced inequality is expected to change social norms and enhance political participation and interest, which consequently could reduce global greenhouse gas emissions.

In an ex-post analysis of a traffic regulation change in Madrid, Perez-Prada and Monzon [33] concluded that a reduced speed limit on a section of the city's 'inner ring', motivated by traffic safety reasons, led to carbon dioxide emission reductions of approximately 15 %. Ribeiro and De Abreu examined four Brazilian initiatives to cut fossil fuel consumption implemented during different periods of time in the transport sector: (1) flexi-fuel technology, (2) the national biodiesel program, (3) the national vehicle efficiency program, and (4) the Rio de Janeiro vehicle inspection and maintenance program [34]. These programs were all motivated by domestic policy goals such as energy efficiency, energy security and job

creation, but yielded total carbon dioxide reductions of approximately 27 Mt CO2e as a co-benefit.

Haley et al. [35] found that if California, which is burdened by recurring droughts, would reach its goal to reduce water consumption by 20% from 2000 to 2020, total greenhouse gas emission savings would amount to 3.5 Mt CO2e in 2020. Similarly, a Danish study concluded that implementing the EU Water Framework Directive could provide synergies in terms of greenhouse gas emission reductions. With the following four measures, the directive could lead to a 35–65 % reduction of total agricultural greenhouse gas emissions within the Roskilde river basin: (1) manure treatment for biogas production and improved nitrogen utilisation, (2) cultivation of perennial energy crops, (3) extensification of intensively farmed lowland areas, and (4) wetland restoration [36].

### 3.3. Meta Studies on Climate-Related Co-Benefits

The growing scientific interest in climate-related co-benefits has led to a range of scientific publications in recent years, and the notion of co-benefits as such has been discussed by some scholars [4]. This surge in co-benefits research has led to a wide use of different conceptualizations, categorisations and definitions, often used interchangeably, which can naturally be confusing. Given the lack of an agreed-upon taxonomy of co-benefits, studies have taken different approaches, with various scopes and inclusion of measures. However, few articles before the review by Karlsson et al. [1] have systematically and comprehensively synthesised these findings. In the following, we present earlier reviews of climate-related co-benefits and how these have been conceptualised and categorised.

In a conceptual review of 138 articles by Mayrhofer and Gupta [5], co-benefits were divided into five categories: 'climate-related', 'economic', 'environmental', 'social', and 'political and institutional'. The structuring of these and where to place a certain co-benefit is, however, imprecise. For example, 'reduced air pollution' and 'improved health' belong to different categories despite being closely related. Deng et al. [6] provide a bibliographic analysis of 1554 papers on the co-benefits of greenhouse gas mitigation. The authors found that impacts on ecosystems, economic activity, health, air pollution and resource efficiency had received most attention, whereas fewer studies had been conducted on areas such as conflict and disaster resilience, poverty alleviation, energy security, technological spill-over, and innovation and food security. Gao et al. [25] focused on only one of the co-benefits identified by Deng et al. [6] in a systematic review of the literature on the public health co-benefits of greenhouse gas emission reductions. Analysing 36 peer-reviewed papers, they identified climate mitigation measures in five key sectors: energy, transportation, food and agriculture, households and industry and economy, concluding that co-benefits can be substantial and thus highly relevant to policy makers. Nemet et al. [37] reviewed 37 studies on the air quality co-benefits of climate policies, focusing on those that put a monetary value on the co-benefits. The review showed a range of values from USD 2 to 196 per tCO2e, with a mean of 49 USD/tCO2e. This can be compared with the EU ETS price level, which was above 70 EUR/tCO2e throughout November [38]. In a similar study by Chang et al. [39] including more recent studies, the estimated monetary value of improved air quality ranged from 2 to 380 USD/tCO2e, nearly doubling the maximum yield found in Nemet et al. [37]. In a review of the co-benefits from food waste reductions and dietary shifts, Some et al. [40] conclude that such demand-side strategies have more synergies than trade-offs with the Sustainable Development Goals (SDGs). Chatterjee et al. [41] reviewed 52 scientific articles concerning the co-benefits from energy efficiency measures in the EU and South Asian countries, finding that such measures could save billions of dollars annually in energy savings in both regions, amounting to annual energy savings worth USD 13 billion in the EU and USD 26 billion in South Asian countries. In a review of 26 meta-studies on biochar application in agriculture to sequester carbon dioxide, Schmidt et al. [42] found that using biochar in agriculture provides several co-benefits such as improved yields and microbial activity as well as reduced water use.

Ürge-Vorsatz et al. [4] reviewed the research on co-benefits from a broader perspective and argued for using the term 'co-impacts', in order to be more neutral in terms of positive and negative impacts, and because policies are seldom introduced for the purpose of generating co-benefits. They also discussed the degree of intentionality behind the impacts, arguing that less emphasis should be devoted to intention and more focus directed to 'co-impacts'. Consequently, the authors argued that co-impacts should be included in decision-making frameworks such as social CBAs, Integrated Assessment Models and multi-criteria analyses, and that doing so could substantially change the outcome of such frameworks, thus being highly policy-relevant.

As evident from this overview, these meta studies included co-benefits of climate policy and measures that go beyond climate change mitigation. In addition, some studies also covered various measures in other policy fields resulting in greenhouse gas emissions reductions, even though we have not come across any comprehensive peer-reviews with that focus.

### 3.4. A Framework and Taxonomy for Climate-Related Co-Benefits

Given the variety of conceptualizations and methodologies for assessing co-benefits, we provide a novel framework and taxonomy for identifying various co-benefits. This framework and taxonomy aim to provide researchers and policy analysts with a user-friendly framework to identify co-benefits, promote policy dialogues and ultimately to support their inclusion in decision-making frameworks and processes. Earlier scientific research has discussed potential reasons for the negligence of co-benefits in policy making. One reoccurring explanation is the lack of policy integration—that is, taking several goals into account when designing policy [43–46]. A plausible reason to why this is the case is the common fragmentation of institutional regimes, with multiple ministries and agencies working in siloes and dealing with specific policy issues [47,48].

In developing this taxonomy, we expand on the review by Karlsson et al. [1] using the same logic concerning the 'direction' of synergy. This means that co-benefits are categorized based on the rank of intention (what is the main intention or objective of a policy?) and impact (what is the main impact of a policy?) of co-benefits. Against this background, a three-type taxonomy of different types of co-benefits can be organised in a framework in which "climate policy co-benefits" are called 'Type 1', "climate co-benefits" from measures in other policy fields are termed 'Type 2', and co-benefits from policies designed to achieve multiple objectives are referred to as 'Type 3' (see Figure 2). While acknowledging that policy objectives are not always clearly articulated and impacts not always straightforward or comparable, we argue that identifying the intention and impact of co-benefits will become more straightforward as the knowledge, awareness and quantification of co-benefits increases. This three-type taxonomy can support such developments. The intention of this distinction is to assist in structuring research and policy-making and to stimulate the further exploration of co-benefits in both the science and the policy domain.

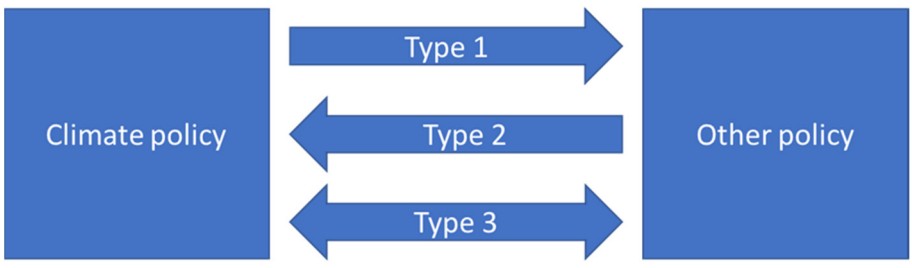

**Figure 2.** A framework and taxonomy for climate-related co-benefits.

While recognizing that this distinction of co-benefits is broad and does not allow for distinctions between sector-specific policy co-benefits, we argue that such a distinction has several merits. Firstly, the wide variety of potential co-benefits resulting from various

policy fields means that it would be practically impossible to provide a taxonomy that describes all distinct and independent co-benefits. This has been noted by other scholars who suggest that a broader categorization of co-benefits could have analytical benefits and spur policy action [4]. A broader distinction based on the direction of the co-benefit's impact, as suggested here, can make it easier for policy analysts and researchers to understand and apply the taxonomy in a variety of different policy contexts. Secondly, a broader distinction makes it easier to compare co-benefits of different policy options within and between policy fields or governmental departments, which is useful for policy makers when making decisions. Others have argued that categorizations of co-benefits should be structured to correlate with governmental departments [49], but given the variety of ways in which governmental departments can be structured and change over time, such categorization would not necessarily be malleable to different policy contexts. A broader and less specific categorization, however, would allow for and promote cross-departmental policy discussions. Finally, a straightforward taxonomy is easier to communicate and could thus stimulate further policy discussions and research. Earlier research has pointed to the necessity of communicating co-benefits, such as Bain et al. [2] who states that "communicating the co-benefits of addressing climate change could provide a way to foster public, and thereby influence government action, even among those unconvinced or unconcerned about climate change" (p. 10). Our categorization offers prospects to spur and incentivize such communication and policy discussions.

While considering these merits, there is no ideal taxonomy, and different taxonomies have different purposes. Given the intention of this taxonomy, we argue that this distinction could improve clarity and promote policy discussions regarding climate-related co-benefits.

With a more structured discussion about climate-related co-benefits, we believe that the importance of co-benefits to policy-making can be better understood and acted upon. Furthermore, and given the wide-ranging use of conceptualizations and categorizations in research, this framework and taxonomy could make research in the field more accessible to policy-makers. In the event that a policy objective is ambiguous, the value of its effects, for example in monetary terms, could decide whether it can be seen as belonging to the climate policy field or some other policy field. This relates to another discussion about the intentionality of "co-effects" [4], and that in an ideal world without compartmentalized policy making and perfect information, every policy measure would be considered and designed for multiple purposes. However, as has been shown in not least Karlsson et al. [1], all relevant facts are seldom accessible.

In the following section, we describe existing and potential climate-related co-benefits and the awareness of said co-benefits in an illustrative case study of Swedish policy-making.

## 4. Climate-Related Co-Benefits in Swedish Policy Making

By applying the co-benefit framework and taxonomy, we have conducted an illustrative study of policy-making in Sweden, a country that has been considered a climate policy leader [10,50–52]—albeit still not reaching its climate objectives [53]. Sweden started to phase out fossil fuels already five decades ago, was among the first countries to tax carbon dioxide and has a high share of renewable energy.

As noted in the methodological section of this paper, the long-standing Swedish Public Inquiry System plays a central role in the policy landscape. Preceding governmental policy proposals and in particular legislative amendments, inquiries as requested by governmental directives are carried out. These inquiries inform the government through reports that investigate specific issues or policy proposals and their socio-economic consequences, often in much detail. All governmental inquiries are regulated by the governmental ordinance on public inquiries [54] and to a lesser extent by the governmental ordinance on impact assessments in legislation [55]. A large number of appointed Inquiry Committees, led by individual investigators or parliamentary balanced committees, produce hundreds of reports for the government annually, often with the ambition to reach broad agreements across the political spectrum in contested issues, not least in the field of environmental

policy. Since governmental inquiries are both detailed and fundamental to the preparatory processes preceding, e.g., legislation, they serve as a good basis for and constitute the main part of our exploration. In some cases, though, we have also studied the eventual governmental bills, as well as specific policy audit reports.

*4.1. Type 1: Climate Policy Co-Benefits*

Swedish policy-making with direct or indirect climate relevance goes back to the beginning of the 1970s, when decisions were taken to decrease the use of fossil fuels, after which a series of laws and fiscal instruments have been adopted [56,57]. However, climate policy aimed at climate mitigation came first with few exceptions after 1990 [58,59]. In the following, a series of key climate policy reforms from 1990 to 2022 in Sweden will be described, starting from the onset of contemporary climate policy after the Climate Convention (UNFCCC) was signed in the early 1990s, through the mid-2000s when more thorough climate policy discussions were held, and culminating in the Swedish climate policy framework in the late 2010s and early 2020s. This exploration is based on a thorough reading and analysis of governmental inquiries of relevance for climate policy carried out during this period, focusing on those that have led to principal proposals to and decisions by the government or parliament.

The starting point for this case study coincides with the adoption of the carbon dioxide tax in Sweden, proposed by the government in 1990 [60]. The proposal followed a governmental inquiry on environmental fiscal instruments [61]. Despite producing over 2000 pages of analysis and reasoning, including pollutants other than carbon dioxide as well, the inquiry did not identify or refer to any co-benefits resulting from its proposal to use an economic tool for mitigating carbon dioxide emissions.

After the UNFCCC was signed and entered into force, a series of governmental inquiries were carried out, most notably from the so-called Climate Delegation established in 1993. In its 1994 report on climate impacts [62], the objective was to stabilize emissions by the year 2000 to 1990 levels and to decrease thereafter, with a focus on cumulative emissions and references to IPCC scenarios [63]. Increased energy and carbon taxation were proposed within the frames of a green tax reform, but nothing was said about co-benefits resulting from the proposed measures. In its second report [64], the delegation elaborated on cost-related issues and on various emissions reduction allocation principles for a UN protocol to the UNFCCC; however, despite obvious co-benefits, these were again absent from the economic analysis. This applies to a number of other climate-focused governmental inquiries, for example the inquiry on joint implementation [65] in its top-down and bottom-up calculated mitigation costs of climate policy, and the inquiry on climate change in transport policy [66], which proposed increased carbon dioxide taxation in parts of the transport sector and elaborated on policy objectives and climate impact. The only exception is the governmental inquiry on so-called "alternative" fuels [67] in 1996, which both included and calculated co-benefits. These were recognized in terms of, e.g., increased employment in rural areas resulting from climate policy measures stimulating renewable fuels, in particular various biofuels. A socio-economic calculation was carried out in which benefits were included and quantified. Parts of this proposal were implemented by the government and the parliament.

Taken together with environmental policy development in other fields, these climate policy proposals led to a broad strengthening of Swedish environmental policy, most notably with the adoption of the 15 environmental quality objectives in 1998 [68]. The eventual 16 environmental quality objectives include specifications and milestone targets for different environmental objectives, e.g., "clean air" and "sustainable forests". Despite comprising and assessing a series of interrelated environmental objectives, co-benefits were, again, not mentioned.

In the first decade after the turn of the millennium, the IPCC produced major assessments in 2001 and 2007 [69,70]. In these years, impact assessments became more common in governmental inquiries in Sweden. Thorough impact assessments were undertaken

in, e.g., the "Long-term inquiry" [71], a recurring study in Sweden, but the analysis of the policy proposals mainly concerned policy costs, costs of environmental damage and consequences for the competitiveness of the Swedish industry. In a short period of time thereafter, following the ratification of the UN Kyoto Protocol, several governmental inquiries [72–75] explored climate policy development and undertook cost-benefit analyses of various proposals, although almost exclusively with an emphasis on costs rather than benefits. The subsequent parliamentary balanced Climate Committee [58] presented a number of policy proposals in broad political agreement and calculated costs and benefits, but it overlooked co-benefits, despite a number of such being obvious, e.g., health benefits from increased fuel taxation and economic savings from investments in efficiency measures in buildings. In the governments' bill [76] following the report from the inquiry, discussions on synergies and benefits were almost completely absent. Instead, the impact assessment presented almost exclusively cost-reasoning concerns. That was also the case in an inquiry on the taxes on fluorinated greenhouse gases [77].

After 2010, the main step in Swedish climate policy was taken based on an inquiry proposal from the governmental All-Party Committee on Environmental Objectives [78]. The committee was charged with developing and analysing a new climate policy framework for Sweden, including stricter long-term objectives and a new climate act, subsequently implemented in 2018 [79]. In parallel, in response to a governmental directive, the Committee also developed a series of proposals aiming to improve air quality [80]. Despite having access to monetary estimates of health costs due to air pollutants and clearly expressing that these costs would decrease substantially with stricter climate policy, the Committee did not include the value of air quality improvements in the estimates of the costs and benefits of its own climate policy proposal. In fact, the Committee even stated explicitly that co-benefits were not accounted for in the quantitative analyses of its proposed policies, despite a background report to the inquiry underlining the missed potential when overlooking co-benefits [21]. Another background report by Klevnäs et al. [81] analysed the general equilibrium and energy system models used to assess the economic consequences of reaching Sweden's climate objective of net-zero emissions by 2045, highlighting that these estimates strongly differ depending on the assumptions and considerations made in the models. An illustrative example of the complexities with modelling was the scenario projections of emissions in the transportation sector. In the scenario assuming a rapid transition to biofuels and electrification, emissions are reduced quickly at low costs. In the slower transition scenario, emphasis is rather directed towards reducing transportation work itself, leading to the high costs of reaching climate objectives. Klevnäs et al. argues that this highlights the difficulties of forecasting future technological development, and that such modelling is bound by uncertainty, potentially creating imbalances in the impact assessments. These imbalances are commonly worsened when co-benefits are unaccounted for, and the authors argue that co-benefits are of particular value in the impact assessments of Swedish climate objectives.

In 2020, an inquiry on climate policy pathways [82] was charged with the task of proposing a strategy for complementary climate actions, e.g., Carbon Dioxide Removal (CDR) technologies and practices, to reach the objective of net-zero emissions by 2045, which was adopted in the referred climate policy framework. In its report, the inquiry states that synergies and co-benefits from complementary actions are significant in striving to reach other environmental objectives, e.g., Sweden's 16 environmental quality objectives. Despite this, there is only a brief qualitative discussion on co-benefits in the impact assessment, and the consequence assessment mainly concerns climate policy costs. Although different co-benefits from CDR technologies are obvious, especially concerning biodiversity [83,84], the inquiry explicitly states that "the cost estimates do not consider, or put value on, other simultaneous benefits that often exist" [82] (p. 737).

During the last two years, additional inquiries have been finalised but so far not translated into governmental or parliamentary decisions. Among these, the inquiry concerning the phasing out of fossil fuels in the transport sector [85] refers to both Type 1

and Type 2 co-benefits in the former case, e.g., in terms of co-benefits following increased use of biogas. However, despite stating that neglecting co-benefits "risk to decrease the socioeconomic efficiency of the proposed strategy" (p. 73), the impact assessment indeed does so. Finally, in 2022, the All-Party Committee on Environmental Objectives presented yet another inquiry report on Sweden's global climate footprint, including a proposal of the consumption-based accounting of greenhouse gas emissions [86]. The Committee again recognised the existence of several "very significant co-benefits" (p. 667), such as reduced emissions of particulate matter due to increased fossil-free transportation and reduced risk of premature deaths, cardiovascular diseases and type 2 diabetes due to increased active transportation, but stated again that these are difficult to calculate. Consequently, they did not carry out any calculations, in contrast with figures presented for several policy costs.

### 4.2. Type 2: Climate Co-Benefits and Tax Policy

The notion of Type 2 co-benefits implies that a decision in any policy field is a potential opportunity for climate mitigation—if not by reducing greenhouse gas emissions, then at least by promoting the avoidance of increased emissions. In this article, we delimit the investigation of policy processes to the field of taxation and a number of key reforms during the study period. Tax policy is chosen since it is highly influential on societal development and on the behaviour of firms and individuals. The main aim of taxation is to finance public sector spending, but taxation also has distributional and business cycle stabilizing and behavioural purposes [87]. The latter is of relevance for internalising external costs, but since any co-benefit of climate taxation would be of Type 1, we do not include such reforms in this section.

The question of which tax reforms have been most important since 1990 is to some extent normative. We have decided to study the reforms pointed out in three comprehensive reports on Swedish tax policy during the relevant period: a review of Swedish tax policy 1992–2009 by the Swedish National Audit Office [88], a paper on tax policy and tax principles in Sweden in 1902–2016 [89], and a recent proposal for a Swedish tax overhaul that contains a historic oversight [87]. For each of the selected reforms, we have studied the governmental inquiry (if there is one), possible public consultancies and evaluation reports, and the government's bill in question.

First, several changes to corporate taxes were implemented in 1994, focusing on reductions and specifically the previous double taxation of owners. The primary aim was to increase company growth, with a specific focus on small and medium-sized companies [90]. We have not found any material behind the reform that mentions any impact on greenhouse gas emissions.

Two years later, in 1996, the VAT on food was lowered. The reform, not preceded by a public inquiry, was motivated as support for low-income households [91]. We have not been able to find any reference to the effects on greenhouse gas emissions in the policy-making process [92]. Furthermore, in a later governmental inquiry on VAT [93], a full chapter discusses VAT on food without mentioning climate effects.

The reduction of the employer fee in 1997 was also not preceded by a governmental inquiry, and similarly to previous tax inquiries, no mention of effects on greenhouse gas emissions have been found in the related documents [94].

Among the main reforms of the governments in office during 2006–2014 was the so-called earned income tax credits (EITC), aimed at reducing taxes for the working population. The EITC was introduced in 2007 and developed in 2008, 2009, 2010, 2014 and 2019. The aim was to provide stronger incentives to work and thus to increase employment. When the EITC was developed in 2009, it was also explicitly motivated as a way to boost consumption, but the bill makes no reference to possible impacts on greenhouse gas emissions [76]. However, a parliamentary debate on the topic in 2011 shows that critiques of this omission did exist and that income tax cuts risk increasing emissions due to growing household consumption. The governments' response, though, implies an understanding that Swedish greenhouse gas emissions instead should be steered through the Swedish carbon tax, and

that emissions based on the consumption of imported goods are the responsibility of the producing country [95].

Two other reforms in the same period were an abolishment of the wealth tax in 2007 [96] and an abolishment of the real estate tax in 2008 (the latter being replaced with a fee and some tax changes related to real estate sales) [97]. Neither of the two bills on the amendments discussed possible impacts on greenhouse gas emissions.

Spanning from 1993 to 2008, a series of tax deductions for renovation (known as ROTs) and household services (RUTs) had been implemented with the aim to increase employment, reduce the share of unreported and untaxed jobs and incentivize those who clean or renovate their homes to instead buy these services and use the saved time for professional work. ROTs have been implemented and abolished several times, whereas a RUT was implemented in 2007. The possible climate impacts of these reforms are not mentioned in any of the bills for or evaluations of the ROT and RUT deductions [98,99]. For the ROT, the Swedish Energy Agency had proposed a specific requirement on energy efficiency. The government commented on this in the bill, stating that while energy efficiency is important, the primary motive for the ROT was to boost employment and reduce the share of unreported and untaxed jobs in the sector; therefore, the proposal was left without further consideration [99] (pp. 32–33).

In summary, none of the studied tax reforms were preceded by any assessment of the potential effects on greenhouse gas emissions. It is not necessarily the case that such effects have existed, yet much speaks for a potential to at least fine-tune tax amendments to achieve emission reductions. Nevertheless, the referred cases illustrate that possible Type 2 co-benefits are unlikely to be identified in Swedish policy-making since they are commonly exempted from analysis.

These findings are hardly unique for the tax policy field. A similar pattern has been shown in a report published by the Swedish governmental inquiry 'Fossil Free Sweden' [100], which, besides taxation, also provided examples from the fields of digitalisation and defence policy, where potential greenhouse gas emissions reductions were generally overlooked [3].

## 5. Discussion

As has been shown in the Swedish case study, climate-related co-benefits are rarely considered in policy making. This strengthens the conclusion in the review by Karlsson et al. [1] that few studies find evidence of co-benefits being acknowledged in policy making. Plausibly, there are several explanations for this negligence, depending on the context in which policy making occurs and the actors involved. In the following, we discuss potential reasons for why co-benefits are seldom recognized in policy making, followed by a more specific explanation in the Swedish case, concluding with a number of recommendations for both research and policy.

It is clear that synergies within environmental policy and between policy fields have long been strived for. In for example the EU, environmental policy integration has been expressed in treaties in force since 1998 [101]. On the international level, Agenda 21 from the 1992 Rio Conference was an early example of a program promoting holistic strategies [102]. Presently, Agenda 2030 and the 17 SDGs play a similar role. Despite these ambitions, policy making seldom considers integration and coherence between environmental and other policy domains [103,104]. Challenges remain, not least regarding climate policy integration [105], with in particular mutual benefits seldom being considered in the economic analyses of policy proposals [3]. To a large extent, this is explained by economics, despite a basically broad societal approach being quite narrow in practice [21–23]. Commonly used economic tools in policy making include computable general equilibrium models that simulate economy-wide impacts of, e.g., climate policy instruments, and social cost–benefit analyses (CBA), which aim to calculate the net social benefits of proposed policies. These appraisal tools ought to include, as far as possible, all relevant consequences of policy alternatives as well as provide alternative scenarios, but this is rarely the case in practice;

both approaches suffer from numerous limitations, including a difficulty to cope with uncertainties in complex socio-ecological systems [22,23,106]. Environmental costs are consequently often underestimated, and broader benefits associated with new policies are commonly overlooked with these appraisal methods [3,21,107,108], for which they have faced strong criticism [21,22,109]. Our analysis highlights that the strong adherence to using such instruments partly explains why policymakers generally neglect co-benefits in decision-making processes. Considering these challenges and biases in contemporary economic analysis, it is unsurprising that co-benefits are commonly omitted in policy-making processes [3,21].

The Swedish case offers a practical illustration of the emphasis on costs rather than benefits. The governmental ordinance on public inquiries [54], which defines what their reports should include, stipulates the following topics as mandatory to assess when, e.g., new legislation is proposed: costs and revenues for the public and private sector; socio-economic consequences in general; the municipal autonomy; crime and crime prevention; employment; conditions for small companies; equality between women and men; and integration.

Strikingly, neither the parliamentary-agreed-upon environmental quality objectives in general nor the oft-prioritised climate objective in particular is on the list. Hence, not even inquiries on energy and traffic systems are obliged to consider how proposals affect emissions of greenhouse gases (unless explicitly asked to do so in the associated governmental inquiry directive). The fact that stipulations for how inquiries should be conducted and what topics to include do not include co-benefits or climate considerations is a plausible reason for their negligence in policy-making processes. Moreover, as has been shown in this study, the socio-economic analysis is predominantly focused on costs rather than benefits. However, a detailed study of the All-Party Committee on Environmental Objectives' process in developing its proposal for Sweden's Climate Act [8] shows that joint learning on complex climate issues in the Committee, as well as continuous expertise consultations and broad deliberations across political blocs, allowed the Committee to put the conventional economic analyses in a broader context. This unveils opportunities in policy-making processes in contexts other than that of the Swedish one to also better consider co-benefits in the future.

Based on the findings of this study and earlier scientific literature, we present a number of recommendations for each type of co-benefit that could incentivize future research and promote the inclusion of climate-related co-benefits in policy making.

### 5.1. Type 1

We argue that Type 1 co-benefits should always have a place in climate policymaking. Many Type 1 co-benefits are well understood and thoroughly researched, and methods often exist for calculating their monetary value.

Here, a key challenge for researchers is to make existing data more accessible for policy makers [26,110,111]. One way of doing that is to present results in a way that can easily be implemented in policy making, such as valuations in USD/tCO2e. If monetary values are unavailable, other quantified data as well as qualitative information should be considered. Furthermore, methods should be developed that enable the transferring of existing co-benefit valuations to new geographies and contexts. In parallel, studies are needed on additional Type 1 co-benefits. More broadly, researchers should also help in developing assessment models and approaches for more comprehensive policy assessments and policy making [112].

Concerning policy makers, the main challenge at present is to use the available data. Governance reforms are also needed to ensure future evidence-based policy making, in which relevant co-benefits are considered [1]. A central step concerning Type 1 co-benefits is to require their inclusion in CBAs when preparing policy proposals. In the Swedish case, adding climate impacts to the referred list of mandatory topics that inquiries should assess would ensure that policy makers are provided with information to consider, to a greater



extent, climate-related co-benefits in all policy areas. Requiring significant emission effects to be monetised would also improve the stipulated analysis of "socio-economic consequences".

### 5.2. Type 2

While the climate costs of emissions resulting from policies on industrial and infrastructural development are often considered nowadays, the potential Type 2 co-benefits are, as we have shown in this article, seldom assessed—not even in more obvious cases such as energy security policy. In seemingly more distant policy areas, such as taxation (outside climate policy), we have not found any evidence in the analysed policy documents of Type 2 co-benefits being considered. Given the referred growing scientific evidence of the existence of Type 2 co-benefits, overlooking them may lead policymakers to make decisions that are socio-economically suboptimal.

Among the key challenges for researchers is to identify possible Type 2 co-benefits within separate policy fields, such as transport, energy, digitalisation, defence and taxation. Results should preferably be quantified and ideally translated into monetary values, such as USD/tCO2e. Similar to the recommendations for Type 1 co-benefits, findings ought to be presented in a manner adapted to fit policy-making requirements.

Compared to the situation with Type 1 co-benefits, policy makers encounter greater challenges with Type 2 co-benefits due to a considerable lack of knowledge and data. Understanding where co-benefits may exist and how large they are might therefore necessitate allocating research funding to the field.

### 5.3. Type 3

Considering different central societal objectives in parallel when developing public policy is intuitively valuable and would require assessing all significant foreseeable effects, including co-benefits. However, since both Type 1 and Type 2 co-benefits, as we have seen, are inadequately considered in science and policy, identifying Type 3 co-benefits is highly challenging. Neither research nor policy making is presently structured sufficiently well to broadly provide the knowledge and data needed to promote Type 3 co-benefits, or what has been called a 'co-impacts' approach [4].

To move in this direction, researchers should strive to generate more data on Type 1 and Type 2 co-benefits and develop models for more comprehensive impact analysis [22]. With the development of increasingly complex models explaining climate change–economy interactions, co-benefits can be added and better understood. (See, e.g., Ref. [113] for an example where reductions of nitrogen oxides pollution are added to the so-called DICE model of climate change and the economy.) Despite being a Type 1 co-benefits study, this kind of model development simplifies the understanding of multiple impacts of policy alternatives.

For policy makers, collaboration across different sectors seems necessary to improve the understanding of all significant effects of different policies. Increasing efforts to apply more holistic and integrated approaches in policy-making processes are thus important for example when it comes to impact assessment requirements. When assigning public bodies, whether permanent or temporary, the task to consider the significant climate-related co-benefits in policy-making would be desirable. In the long-term perspective, new comprehensive assessment approaches are needed, but since practical and streamlined methods are still immature in both research and policy, traditional assessment methods ought to be adapted to increasingly consider Type 3 co-benefits.

### 6. Conclusions

This article has assessed the scientific literature on climate-related co-benefits and has developed and applied a novel co-benefits taxonomy and framework in a concrete policy setting. The extensive analysis of state inquiries and related policy documents and policy reforms highlights that co-benefits are seldom recognized in either preparatory stages of policy-making nor reflected in subsequent policies. Given the extent and significance of

various climate-related co-benefits, this must be considered a serious flaw. We have argued that the proposed taxonomy and framework could incentivize future research and spur policy dialogue on co-benefits. In that respect, we have provided a set of recommendations for both research and policy that could incentivize well-needed and urgent avenues for future research and policy development on climate-related co-benefits.

Identifying, assessing and considering co-benefits during policy-making would help to prevent biased and suboptimal outcomes. In relation to adopted climate policy objectives, acknowledging and applying co-benefit values in models and processes would clarify the need for more ambitious emissions reductions, not least given the findings of the IPCC's sixth assessment reports [7,114,115] and the fact that global greenhouse gas emissions are still rising [116]. We argue that the presented taxonomy and framework can help policy analysts and policy makers to structure their understanding and operations. To achieve this, we see a need to reform policy-making processes and decision-making criteria.

Concerning policy-making processes, the setup ought to reflect the multiple dimensions and complexities often involved when dealing with climate change issues. Ensuring that multiple competences are continuously involved in policy making—for example in expert and political committees in agencies, governmental ministries and parliaments—is key. Expertise in economics is far from sufficient.

Turning to decision-making criteria in agencies, preparatory committees and inquiries, and legislative processes as such, the type of evidence that is called for, as well as how uncertainty is assessed, influence what evidence that is searched for. Adding decision-making criteria stipulating that co-benefits are considered would strengthen the evidence base, help to reduce uncertainty and aid in more optimally balancing between all known relevant costs and benefits in decision making.

**Supplementary Materials:** The following supporting information can be downloaded at: https://www.mdpi.com/article/10.3390/cli11020040/s1, Figure S1: Based on Scopus and the following search string used on November 30, 2022: (TITLE-ABS-KEY (benefit AND "climate policy")) OR (TITLE-ABS-KEY (co-benefit OR cobenefit AND climate)) OR (TITLE-ABS-KEY ("ancillary benefits" AND climate)) OR (TITLE-ABS-KEY ( "double dividend" AND climate)) OR (TITLE-ABS-KEY ("win-win" AND climate)) OR (TITLE-ABS-KEY (synergy AND "climate policy")) AND (LIMIT-TO (SRCTYPE, "j")) AND ( LIMIT-TO (LANGUAGE, "English")) AND (EXCLUDE (PUBYEAR, 2023) OR EXCLUDE (PUBYEAR, 2022)).

**Author Contributions:** Conceptualization, M.K. and N.W.; methodology, M.K. and N.W.; validation, M.K., N.W. and O.L.; formal analysis, M.K., N.W. and O.L.; investigation, M.K, N.W. and O.L.; writing—original draft preparation, M.K. and N.W.; writing—review and editing, M.K., N.W. and O.L.; visualization, M.K. and N.W.; project administration, M.K.; funding acquisition, M.K. All authors have read and agreed to the published version of the manuscript.

**Funding:** This work was originally based on support by Formas, The Swedish Research Council for Environment, Agricultural Sciences and Spatial Planning (grant number 2017-01877 for the project 'Policy Synergies and Climate Change').

**Data Availability Statement:** Not applicable.

**Acknowledgments:** Eva Alfredsson participated in the project, co-authored Karlsson et al. (2020) and has contributed in discussions on the proposed taxonomy, but she has not participated in writing this specific article.

**Conflicts of Interest:** The authors declare no conflict of interest.

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
