# Peer review of "Climate-Related Co-Benefits and the Case of Swedish Policy"

_climate, doi:10.3390/cli11020040_

Round 1

Reviewer 1 Report

Some of the most significant issues facing humanity today are climate change, especially global warming. Because of their role in generating greenhouse gases (carbon dioxide, methane, nitrous oxide, and fluorocarbons), which act as a heat trap in the atmosphere, human activities are mostly to blame for global warming. Seeing the disaster associated with it any constructive attempt to reduce its impact would be of paramount importance. Neglecting such attempts may attract risk. In this paper, the Swedish case study has demonstrated how rarely climate-related co-benefits are taken into account when formulating policies. I found it interesting. This may help readers to understand the framework which may help policy analysts and policy-makers. Good to learn while framing policies. This paper may be published. 

Reviewer 2 Report

The manuscript proposes a framework (taxonomy) concerning climate-related policy co-benefits. The taxonomy involves three types of co-benefits: climate policy co-benefits (e.g., improved air quality), co-benefits for climate objectives from policymaking in fields other than climate (e.g., taxation), and co-benefits from policies designed to achieve multiple objectives. The proposed framework is discussed using the case of Sweden as an example. 

The Introduction section is very concise. Although the research motivation is stated, it is not sufficiently elaborated. The main justification for the scope of the manuscript relies on a single comprehensive review article. There are no concrete examples of, for instance, “…the societal opportunities linked to climate mitigation…” or of synergies concerning “…policies and measures in other fields benefit climate objectives…”. 

The manuscript does not involve a Methodology section, which is a major shortcoming.

The discussion in lines 60-65 does not clearly explain why policy objectives are not always articulated. A better statement of this discussion is needed.

Lines 69-70: It is not clear what incorporating co-benefits in economic analysis of climate change policy refers to, how this can be achieved, and what contributions such analysis could bring.   

Since it is not explicitly defined and discussed, there is a confusion about what the manuscript seeks to identify in terms of co-benefits. It might be researchers acknowledging/identifying/ discussing/analysing different types/categories/areas of co-benefits pertaining to policies/implementations/strategies/actions. Which one(s) of these are within the focus of the manuscript is not explained. The text switches between, for instance, acknowledging the co-benefits by researchers and the discussion of the co-benefits themselves as the focus. 

It is also not clear whether the focus is solely on policies. If this is the case, it needs to be stated and also reflected in the title of the manuscript.   

Lines 104-105: It would be better to discuss the types of analysis and findings presented in the articles that employ quantitative methods. 

Line 118: In which sense are policies and measures in other fields contrary to climate policy?

Lines 154-156: What are the disadvantages associated with different conceptualisations, categorisations and definitions concerning co-benefits?

Lines 206-218: The framework presented in Section 2.4 (Figure 1) is key to the manuscript since it defines the proposed taxonomy and the proposed means for analysing co-benefits. It seems that the framework is based on the assumption that a set of policies can be identified that solely target greenhouse gas mitigation (Type 1) but do involve co-benefits associated with ‘other policies’. What does not seem to be explained rigorously is how this assumption is made and how valid it is. On a related issue, the terms ‘other policies’ or ‘policy fields’ may also be problematic because no evidence is presented to support the idea that policymaking is often compartmentalised. On the contrary, policies may cover a spectrum of fields/topics. Also, it is still not clear which conceptualisation is being discussed/criticised. Is it the conceptualisation by researchers, policymakers, or by other stakeholders? It is important to note that the answer to this question has different types and magnitudes of impacts.  

The presentation of the framework presented in Section 2.4 remains insufficient and does not provide evidence regarding why it is novel and what contribution it provides. With the current discussion, the framework can be understood as simply defining co-benefits that originate from climate, non-climate and mixed-objective policies.    

Line 243: Has sufficient information built-up for evaluating policies implemented in September 2022?

Lines 273-279: It would be better to provide references to support the discussion, especially the first sentence. 

Line 285: Do environmental fiscal policies fall into the Type 1 category with respect to the proposed framework? Could they also be perceived as Type 2 (as referred to ‘taxation’ in Line 13)? How is this distinction made? This needs a more clear explanation in Section 3.1, where environmental fiscal policies are mentioned, or in Section 2.4, where the framework is defined. 

Similar comments also hold for Section 3.2, where Type 2 policies are discussed. 

The discussion in Sections 3.1 and 3.2 is difficult to follow and involves a number of assessments that are not supported by evidence from the literature or other sources (policy documents, laws, directives).

Lines 464-466: It is stated that “…the potential reasons for why co-benefits are seldom recognized in policymaking…” will be discussed, however, the discussion does not highlight these reasons. It would be much better to provide a more item by item discussion of the potential reasons. 

It is not clear how the recommendations in Sections 4.1-4.4 are derived (e.g, the sentence in Lines 539-540: “In the case such [Type 2] benefits are overlooked, policies have not been socioeconomically optimal.”). Considering that there is no earlier discussion on Type 3 benefits, this is even more difficult to comprehend. A similar conclusion holds for governance reforms discussed in Section 4.4. This is not to say that the recommendations are not viable. However, as the authors state the aims and the contribution of the manuscript, it needs to be based on a framework and a related flow of ideas rather than an opinion article on policies and policy recommendations.   

The manuscript does not involve a Conclusion that would summarize and wrap up the discussion. 

Round 2

Reviewer 2 Report

The revised version of the manuscript is considerably improved.

The Introduction section is extended to include a much clearer statement of the aim of the manuscript. However, as highlighted in the first review, explicitly discussed examples of co-benefits are still missing.

A Materials and Methods section is also added. However, the “inductive exploratory approach” stated as the methodology for the first part of the manuscript needs to be better explained, including a discussion of the methodology (e.g., objectives, suitability, advantages, drawbacks) and references to earlier studies using the methodology. Likewise, the “case study” approach utilized in the second part of the manuscript needs to be further discussed and justified.

Concerning quantitative policy analysis, the manuscript only states what studies utilizing such methods do not consider or discuss. For a more plausible discussion, it would be better to

The disadvantages associated with different conceptualisations, categorisations and definitions concerning co-benefits are now briefly discussed in the revised manuscript. However, since this discussion would highlight the proposed framework's contribution, it needs to be extended. 

The framework and the justification for the taxonomy are now much better explained.

Since there is no sufficient information for evaluating policies implemented in, for instance, September 2022, it could be better to limit the timeline of analysis for the case study or explain how these policies are evaluated.

A Conclusion section is now added. However, it needs to be extended, at least, by including a discussion on if and how the objectives of the manuscript (as stated in the Introduction section) are achieved, directions for future research, and how the findings of the manuscript can be utilized (e.g., for policy making). 
